# Brand Personality as a Consistency Factor in the Pillars of CSR Management in the New Normal

**Elizabeth Emperatriz García-Salirrosas** [1,2] and **Javier Mayorga Gordillo** [3,*]

1    Department of Humanities, Faculty of Business Studies, Universidad Privada del Norte, Lima 15314, Peru; elizabeth.garcia@upn.edu.pe
2    Faculty of Engineering and Management, Professional School of Business Administration, Universidad Nacional Tecnológica de Lima Sur, Lima 15816, Peru
3    Marketing Department, Management Sciences Faculty, Universidad Autónoma de Occidente, Cali 760030, Colombia
*    Correspondence: jmayorga@uao.edu.co

**Abstract:** During such a complex crisis as the one experienced by humanity since the first quarter of 2020, it is necessary to develop tools that contribute to creating the corporate image for organizations that are currently interested in being identified as brands with high social and environmental commitment. Likewise, elements that contribute to building strong brands during a context that has changed consumption priorities are required. For this reason, this paper aims at adapting the dimension of socially responsible brand personality (SRBP), proposed by Mayorga (2017), taking the situation experienced due to the COVID-19 pandemic as a new context. The objective of this research is to contribute to the management of corporate social responsibility (CSR) by providing, from a communicative perspective, a tool that optimizes the creation of a socially responsible image by the different stakeholders. The results allow us to conclude that there is a structural modification of the brand personality proposed by Mayorga, which can be presumed to be generated by the current environment, and which, therefore, can be established as a pillar of CSR management in the new normal, from a relational point of view. The findings clearly identify the virtue of integrity in brand personality, which is made up of two attributes, which, in turn, are made up of 17 traits that can identify a socially responsible brand.

**Keywords:** brand personality; corporate social responsibility; CSR management; branding; sustainability

## 1. Introduction

The brand is considered one of the most valuable intangible assets that companies have, so brand management has become a managerial priority in recent times [1]. One of the important functions that brands fulfill is related to their influence on the buying decision-making [2]. This is due to its leading role in the identification and differentiation of products or services in the market [3]. Its importance lies with the positioning in customers' minds; it is a key aspect to differentiate and establish a competitive advantage [1]. It is worth saying that brands represent perceptions and emotions of customers regarding goods or services of organizations, so that those that are successful in the market establish deep connections with their consumers [4]

The context of universal crisis caused by the COVID-19 pandemic, characterized by uncertainty and the feeling of constant risk, has led to a drastic change in the way consumers behave [5] turning this situation into a challenge for both managers and scholars to seek new ways to deal with this situation [6]. This pandemic will bring profound changes regarding sustainability [7], which means a new beginning for sustainable consumption [8], where consumers favor the consumption of products and services of companies that demonstrate responsible behavior with their different stakeholders [9]. Due to this new

reality, companies see the need to innovate in order to change themselves and implement new forms of management, from a perspective that is more committed to their environment.

One of the practices in business management that organizations have been implementing in recent years, in response to the socio-environmental demands of different stakeholders and as a strategy for their competitiveness in the market, has been the adoption of corporate social responsibility (CSR), which, according to [10], consists of a voluntary commitment of the organizations to generate added value to society through their business activities and, thus, seek the triple impact, which is as follows: economic, social, and environmental, contributing to sustainable development [11]. For this, companies have had to apply policies and systems that benefit their various stakeholders [12–14]. This implies not only rethinking its production and commercial processes, but also its strategic business objectives [15]. These change efforts are important for companies, since they allow, among other benefits, the development of intangible assets, such as the reputation and corporate image of their organizations [16,17].

Corporate image is strongly related to brand equity and this is, in turn, related to its physical and behavioral attributes [18]. According to Keller and Lehman [1], these attributes, values, and benefits that the public identifies in a brand are generated by their perception of it. Based on this and from corporate communication, mainly from communicative strategic thinking, it is important to propose tools that help build the corporate identity of companies, such as a brand personality.

The marketing view of brand personality has been studied and debated by several authors [19–26] establishing in almost all of these studies that consumers give human characteristics to brands in order to generate mental associations with them and, in this way, to make a relationship with them and make the buying decision.

For research in corporate communication, marketing and strategic management, the study by [26] has become a tool at the strategic level that helps the company to meet its objectives. For this, it has been replicated in different countries, cultures and contexts, seeking to adjust the original proposal and relating it to different industries. In addition, it has been referenced, refuted and criticized by many other authors in their articles and proposals [27–42].

As the brand is such an important element for organizations and taking into account the context of society, which demands more responsible behavior, tools that contribute to the management of the brand are necessary in order to concentrate communication and marketing efforts in the effective dissemination of CSR actions to enhance brand equity in the market [43]. For this, [44] created an instrument called the "Instrument for the Assessment of Attributes that define the Socially Responsible Brand Personality (VAR-SR)", which structures a sixth dimension to the model proposed by [26] mainly focused on the characteristics of a socially responsible brand personality (SRBP), which motivates companies to become agents of social change and position their brand strategically facing the changing and competitive environment, whose stakeholders seek sustainable development.

Although there are different studies on brand personality and [44] has allowed us to improve our knowledge about the structure of a socially responsible brand (SRB), the data on which this study has been based on had contexts prior to the so complex global crisis that humanity has been experiencing since the first quarter of 2020, so it is necessary to update the tools that contribute to creating and managing the corporate image of organizations interested in being identified as brands with high social and environmental commitment, as a business strategy to remain in the market in the new normal.

For this reason, this paper aims at adapting the dimension of the socially responsible brand personality (SRBP) proposed by [44], taking the situation experienced due to the COVID-19 pandemic as a new context. The objective of this research is to contribute to the CSR management by providing, from a communicative perspective, a tool that optimizes the creation of a socially responsible image by the different stakeholders.

Under this approach, this study aims at contributing to the management of the brand, from a sustainable perspective, in the new normal. Therefore, the findings of this research

allow us to identify a clear structure of a socially responsible brand, which provides the academy with the identification of a theoretical construct to be deeply analyzed, given the current market conditions. In turn, the industry allows identifying elements that direct communication and the relationship of brands with their different stakeholders. It also allows marketing management to structure brands with a high degree of integrity from a social approach, and, finally, it contributes to CSR management, since it allows strategic plans to be supported by attributes clearly identifiable by the stakeholders, in the identity of companies with a high degree of social and environmental awareness.

After the introduction, this article presents a review of literature on corporate social responsibility (CSR), and brand personality as a driver of competitive advantage (Sections 2.1 and 2.2). The reader can find the theoretical framework of this study in Section 2.3. Then, Section 3 describes the methodological framework and data collection procedure. The results and discussion are presented in Sections 4 and 5. The conclusions, implications, limitations, and future research are presented in Section 6.

## 2. Literature Review and Theoretical Framework

### 2.1. Corporate Social Responsibility (CSR), the Cornerstone of Business Sustainability

There is a large increase in literary contributions that show a trend towards conceptual consolidation concerning social, environmental and financial well-being, since society demands a greater responsibility in business activities. Additionally, significant contributions have been made to the theory and application of the concept of corporate social responsibility (CSR) in conjunction with the development and prosperity of society being linked to the business sector [45].

Castellanos et al. [10] argues that CSR consists of a voluntary commitment of the organizations to generate added value for society from their business activities, having a broader vision than only generates economic benefits. That is, the objective of CSR practice is to contribute to sustainable development and achieve the triple impact, as follows: economic, social, and environmental [11]. For this, companies must apply policies and systems that benefit them so much as well as their various stakeholders [12–14].

In this sense, organizations can voluntarily measure the impact of their CSR actions by applying the indicators proposed in the Global Reporting Initiative (GRI) model (2013), whose objective is to promote the preparation of sustainability reports for every kind of company. These reports consist of annual reports that allow the rendering of accounts and disclosure of their results on the organization's performance in the economic, social, and environmental fields, and are available for the stakeholders as well as internationally through virtual platforms.

The incorporation of CSR in the company involves a whole set of phases of change, not only in the rethinking of the production process, but also of strategic business objectives [15]. The benefits that companies can have by adopting good sustainable environmental behavior are a reduction in risks due to socially not responsible behaviors, a sign of good management quality, cost reduction, new business opportunities, and the development of intangible assets, such as reputation, corporate image, among others [16,17].

### 2.2. Brand Personality as a Driver of Competitive Advantage

A brand is an asset that generates great benefits for any organization and increases the competitiveness of companies. It is understood that brand equity is strongly related to corporate image, which is defined as the impression created in the mind of the public about a company, and is related to its physical and behavioral attributes [18]. For this, companies must create and take care of a positive image, since it means the preference of consumers over their competitors [22,26,31,46].

Theoretically, the brand is defined as a sign or name that identifies and differentiates a product or service from the competition [47].The brand is also understood as a guarantee that will affect how consumers perceive the company, since it attributes levels of quality, reliability, use, and consumption to the products [48,49].A relevant characteristic of the

brand is related to the experiences and opinions of consumers, which lead to the act of buying or using a product, and are not necessarily due to a specific attribute or physical benefit of the brand [1]. Hence the relevance of understanding the importance of the consumer's experience with the products or services of the organization.

Brands have a spontaneous and close relationship with consumers, then researchers have related the concept of a brand personality, granting it emotional dimensions similar to those of human beings [22,26]. In this sense, the human personality is taken as a reference to obtain better results on the brand personality, allowing the preparation of practical instruments consisting of the analysis of the following five factors: activity, responsibility, aggressiveness, simplicity, and emotionality, which provide essential information to managers for the consolidation of their brand [50].

The interest in brand personality is very important for senior executives and researchers [25] because it influences the process of identifying products or services, and the buying decision-making [2,3]. Besides, in a complex and challenging context such as the current one, organizations tend to cut investment in advertising and promotion. However, the perceived attractiveness of the brand personality allows it to remain in the consumer's mind until the next promotional cycle returns to reinforce the image [19], that is, it allows to keep customer loyalty] [37].

Escobar-Farfán and Mateluna [31] carried out an analysis of the different models proposed in the literature on brand personality, from 1997 to 2015, finding that the [26] has been validated and analyzed in its five dimensions of personality in different contexts and realities. This also affirms that it is reliable and replicable, so the concept of brand personality is useful and necessary to be constantly investigated in order to confirm its validity and reliability, since the consumer's mind is complex because his/her opinion is dynamic and easily influenced.

### 2.3. Theoretical Framework

The framework of this study is the proposal made by [44], who structured a sixth dimension to the model proposed by [26], but mainly focused on the characteristics of a socially responsible brand personality (SRBP), which motivates companies to become agents of social change and position the brand strategically, facing the changing and competitive environment, whose stakeholders seek sustainable development. As seen in Figure 1, brand personality must become a strategic CSR tool so that in this way relationships with stakeholders are generated, based on the commitment of both the consumers and companies to the present and future society.

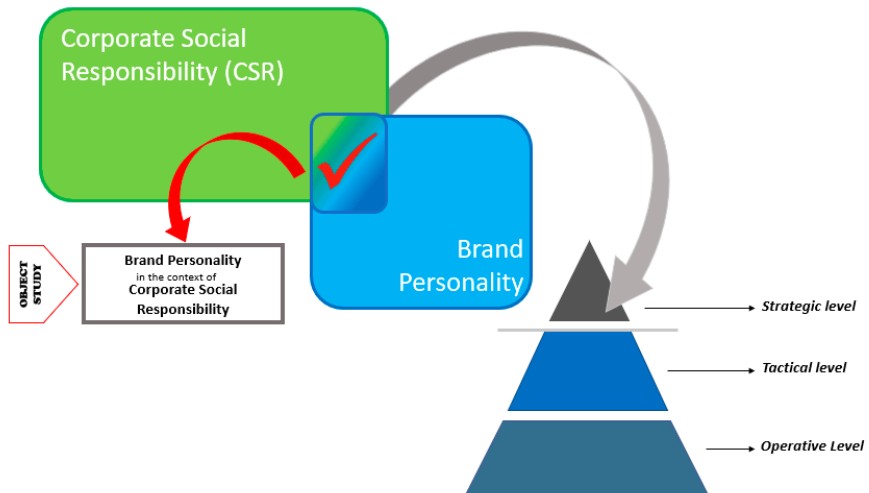

**Figure 1.** Brand personality in the context of CSR, [44].

## 3. Materials and Methods

### 3.1. Instrument for the Assessment of Defining Features of SRBP

The updating and structuring of a socially responsible brand personality, which is the purpose of this work, starts from the consultation of a group of university students about the assessment of the appropriate descriptors to structure it, taking into account that such an assessment process is carried out in a context convulsed by the COVID-19 pandemic.

The data collection instrument used is a version of VAP-SR, a tool originally designed and used by [44] for the constitution of the SRBP. The instrument goes "to the representation of an imaginary subject that is characterized by having a marked socially responsible personality" [44]. It is intended that informants assess the relevance of each adjective assigned as a descriptor of the subject. To be adapted to this research context, VAP-SR had some adjustments, specifically in the description of the imaginary subject and additionally in the items related to demographic information and the performance of social activities, and its application was carried out by using virtual media. The essence of VAP-SR is a list made up of 30 attributes identified by [51].

For data collection, a self-administered questionnaire was distributed through online surveys. These questionnaires were developed using the Google Form, and 980 questionnaires were distributed online, via email, inviting people to participate in the study voluntarily. Responses (502) were obtained, however, after collecting the data, careful scrutiny was carried out to exclude questionnaires that were poorly constructed and the most important questions had not been answered. Finally, 442 responses were considered in the study.

The information obtained from participants becomes the input for the theoretical formulation of the structure of SRBP dimension in a crisis context. The structure obtained from this research allows defining the associations that derive from the traits with the purpose of extrapolating and visualizing the perception of the traits that brands with socially responsible characteristics should have for the public in a convulsive context compared to the study by [44], besides defining stable traits or attributes in both cases.

### 3.2. Participating Subjects

A total of 442 participants from different Latin American countries, although they were mostly young university students from Lima (Peru), who were in preventive lockdown due to the pandemic and took their classes virtually. Table 1 shows the descriptive statistics of all respondents. They were selected by a type of sampling called "Non-probabilistic convenience sampling", according to [52] (p. 230). This type of sampling "allows selecting those accessible cases that agree to be included, based on the convenient accessibility and proximity of the subjects for the researcher".

The proposed statistical analysis revolves around multivariate techniques included in covariance and grouping structure, specifically the factor analysis. According to the recommendations of different authors, a sample size is agreed, taking as a reference what was stated by [53,54], who say that the optimal sample should be greater than 100 subjects and that the minimum acceptable is at least five times the number of variables to be analyzed, but the most acceptable would be 10 times the number of variables to be analyzed, given the type of analysis carried out.

Therefore, as VAP-SR contains 30 essential and 7 complementary variables, and taking into account the authors' recommendations, a minimum sample of 370 subjects is established. Finally, there were 442 informants in the study, that is, 14.7 subjects per essential variable.

**Table 1.** Descriptive statistics of respondents (N = 442).

| Characteristics | Category | Frequency (N) | Percentage (%) |
|---|---|---|---|
| Gender | Female | 277 | 62.7 |
| | Male | 165 | 37.3 |
| Age | 16 to 20 years old | 167 | 37.8 |
| | 21 to 25 years old | 187 | 42.3 |
| | 26 to 30 years old | 56 | 12.7 |
| | 30 to 40 years old | 26 | 5.9 |
| | 41 years and over | 6 | 1.4 |
| Country | Unanswered | 9 | 2.0 |
| | Argentina | 2 | 0.5 |
| | Colombia | 1 | 0.2 |
| | Ecuador | 1 | 0.2 |
| | Peru | 429 | 97.1 |
| Profession | Unanswered | 2 | 0.5 |
| | Business management | 400 | 90.5 |
| | Communication | 1 | 0.2 |
| | Accounting | 4 | 0.9 |
| | Gastronomy | 1 | 0.2 |
| | Engineering | 26 | 5.9 |
| | Nutrition | 1 | 0.2 |
| | Psychology | 4 | 0.9 |
| | Tourism | 3 | 0.7 |

### 3.3. Design of the Research

It is highlighted that this is a replication of the study by [44], therefore the design determined in that research will be followed, that is, a descriptive strategy, since this is a non-experimental study. According to [55], this type of strategy can be observational or elective, given the author's conditions. It is determined that this is a selective study.

The variables that structure the analysis of this research are as follows:

1.  Dependent: valuation variables of the attribute relevance for the description of "socially responsible" individuals, that is, 30 essential variables;
2.  Independent: sociodemographic variables of participating subjects (age, gender, training cycle, place of birth, etc.).

### 3.4. Statistical Methods Used for Analysis

The data analysis was supported by both univariate and multivariate statistical tools, in particular factor analysis and cluster analysis, to respond to the identification of the personality structure of the socially responsible brand, precisely because these techniques allow the grouping of variables and the materialization of latent variables in the data.

On the other hand, in univariate methods, besides basic statistical description (descriptive statistics), the Cramer coefficient is used for the analysis of contingency tables to detect the influence of some sociodemographic aspects on the assessment of adjectives.

These data analysis tools are used in various fields of knowledge for the construction of segments, taxonomic structures, identification of variables not perceived at first sight, among others. Economics, sociology, health sciences, among others, have made use of these tools in their research; in the field of business management, they are mainly used for market segmentation.

## 4. Results

### 4.1. General Analysis of Results

Understanding that human personality is defined in terms of the reactions of individuals to others in different repeated interpersonal situations, a set of characteristics could therefore be established helping to describe a personality with clear propensity to social commitment. In this sense, since 2017, Mayorga and Añaños [56] J. Mayorga [44], and J. Mayorga and Añaños [51] have been working on the structuring of one dimension of SRBP. The authors have presented different documents, where they express in detail the process of dimension definition and design.

In the first proposal of this dimension, it is concluded that

> *"By not including in the dimension the adjectives that evidenced a notable dependence on the personal characteristics of the subjects who valued them, the proposed socially responsible dimension ( . . . ) contains some characters that, when transferred to brand personality, are exempt from contextual influence of audiences."* [44] (p. 272).

The results of this research seek a dimensional structure of a brand personality with high social commitment, being specific for the current pandemic context, updating the one that is currently developed.

### 4.2. Analysis of the Features Defining the SRBP Dimension

In the first stage of the analysis, the assessment degree of each of the traits is determined, which is quantified by using statistical indicators that correspond to the distributions of their frequencies for each of the variables assessed.

Table 2 shows the results of these indicators in each of the adjectives assessed, ordered from the mean obtained, and determine that the adjective with the highest valuation is humanitarian and the one with the lowest valuation is disinterested. The method to know valuation mean starts from the assignment of a numerical value to each answer option, in order to carry out a mathematical calculation.

**Table 2.** Statistical summary of the assessment of adjectives.

| Adjective | Hierarchy | Mean | Standard Deviation | Asymmetry |
|---|---|---|---|---|
| Humanitarian | 1 | 92.443 | 13.782 | −3.171 |
| Solidary | 2 | 91.765 | 14.128 | −2.991 |
| Generous | 3 | 91.199 | 14.821 | −2.659 |
| Collaborative | 4 | 91.018 | 15.448 | −3.115 |
| Helpful | 5 | 90.475 | 16.822 | −2.820 |
| Committed | 6 | 89.751 | 16.432 | −2.652 |
| Empathic | 7 | 89.615 | 17.781 | −2.723 |
| Positive | 8 | 89.186 | 17.070 | −2.615 |
| Responsible | 9 | 88.597 | 17.085 | −2.373 |
| Comprehensive | 10 | 88.439 | 16.456 | −2.227 |
| Kind | 11 | 88.371 | 18.097 | −2.306 |
| Respectful | 12 | 88.032 | 17.129 | −2.319 |
| Charitable | 13 | 87.896 | 18.365 | −1.982 |
| Noble | 14 | 87.511 | 18.532 | −2.024 |
| Optimistic | 15 | 86.900 | 20.251 | −2.413 |
| Enthusiastic | 16 | 86.335 | 19.170 | −1.960 |
| Encouraging | 17 | 85.520 | 20.851 | −2.148 |
| Honest | 18 | 84.819 | 21.415 | −1.940 |
| Integrator | 19 | 84.072 | 20.463 | −1.753 |
| Sincere | 20 | 83.484 | 21.657 | −1.900 |
| Trustworthy | 21 | 83.416 | 21.902 | −1.674 |
| Sensitive | 22 | 82.579 | 21.658 | −1.624 |
| Equitable | 23 | 82.059 | 22.337 | −1.555 |
| Protective | 24 | 81.629 | 23.378 | −1.604 |
| Charismatic | 25 | 78.416 | 25.435 | −1.366 |
| Ecologist | 26 | 77.760 | 26.983 | −1.341 |
| Special | 27 | 74.457 | 26.892 | −1.045 |
| Modest | 28 | 71.878 | 28.790 | −0.933 |
| Tireless | 29 | 65.860 | 29.212 | −0.587 |
| Disinterested | 30 | 36.131 | 42.594 | 0.593 |

Likewise, Table 1 shows the degree of variability in the assessments made of each adjective, taking as an indicator the standard deviation and the bias of frequency distribution, which is calculated on the basis of the asymmetry coefficient and confirms the hierarchy of valuation.

Subsequently, the relationship between the assessment of each adjective and the demographic characteristics of the subjects is analyzed. For this purpose, a statistical independence test is used, quantified through the Cramer contingency coefficient. The purpose of using this statistical technique is to obtain a list of adjectives whose valuation is independent of the demographic characteristics of the informants, and, which in turn is the initial stage for the conformation of the SRBP dimension.

Based on this analysis, the exclusion conditions of the adjectives that will not be part of the final structure of the determined dimension are established. Like [44,56], the choice of adjectives that will be part of the structuring of the SRBP dimension is done by taking as an input the data obtained in the independence test and the mean, dispersion and asymmetry, which are determined as the methods that allow each of them to decide autonomously on the eligibility of the adjective under consideration for the constitution of that dimension.

This work uses a criteria to exclude an adjective from the final choice of the dimension structure; these are the adjectives that, for its assessment, meet at least two of the following conditions (the same ones used by [44,56]). The arithmetic average is less than 16.7 percentile of their distribution;

3.　　Fisher's bias coefficient is higher than 83.3 percentile of their distribution;
4.　　The standard deviation is higher than 83.3 percentile of their distribution;
5.　　The *p-value* of the Cramer test for the independence of attribute assessment with each of the sociodemographic aspects is less than 0.01.

Table 3 shows the results of the assessment analysis of the 30 adjectives in terms of their position, mean, dispersion and coefficient of asymmetry. It also allows observing the results of the analysis of the statistical independence tests to each adjective with each of the dependent variables.

The results allow us to observe that the assessment of the adjectives *disinterested, protective, responsible, comprehensive, generous, kind, positive, trustworthy and solidary* depends statistically ($p < 0.01$) on age. In relation to gender, it is identified that the intensity of statistical dependence is significant ($p < 0.01$) in the attributes *sincere, disinterested, humanitarian, charitable, special, committed, responsible, comprehensive, generous, kind, optimistic, empathic, helpful, solidary, sensitive, equitable and collaborative*.

On the other hand, in relation to the link of the subjects to social activities, the results show that there is a statistically significant relationship of that variable ($p < 0.01$) with the assessment of the attributes *charitable, comprehensive, hopeful, and empathic*. Finally, regarding the profession, it is observed that this variable has a statistically significant relationship with the assessment ($p > 0.01$) of the attributes *respectful, positive, and honest*.

Finally, after counting the mentions of exclusion conditions presented by each of the attributes, the list of adjectives that will be part of the final structure of the SRBP dimension is established, leaving 13 adjectives excluded. That list is made up of the following 17 attributes: *sincere, humanitarian, protective, committed, integrator, enthusiastic, encouraging, noble, respectful, optimistic, helpful, trustworthy, honest, charismatic, sensitive, equitable and collaborative*.

**Table 3.** Summary of the result of the adjective exclusion method.

| Adjective | Profession Cramer's V | Profession p-Value | Age Cramer's V | Age p-Value | Gender Cramer's V | Gender p-Value | Realization of Social Act Cramer's V | Realization of Social Act p-Value | Media | Standard Deviation | Asymmetry | Number of Exclusion Criteria |
|---|---|---|---|---|---|---|---|---|---|---|---|---|
| Sincere | 0.084 | 0.6197 | 0.1066 | 0.2616 | 0.1735 | 0.0099 | 0.1576 | 0.0268 | 83.484 | 21.657 | −1.900 | 1 |
| Disinterested | 0.130 | 0.0584 | 0.1581 | 0.0048 | 0.2219 | 0.0002 | 0.1241 | 0.1466 | 36.131 | 42.594 | 0.593 | 5 |
| Humanitarian | 0.098 | 0.2095 | 0.1242 | 0.0340 | 0.2131 | 0.0002 | 0.1359 | 0.0428 | 92.443 | 13.782 | −3.171 | 1 |
| Charitable | 0.102 | 0.1632 | 0.1354 | 0.0127 | 0.2514 | <0.001 | 0.1873 | 0.0014 | 87.896 | 18.365 | −1.982 | 2 |
| Protective | 0.138 | 0.0331 | 0.1513 | 0.0095 | 0.1326 | 0.1005 | 0.1305 | 0.1107 | 81.629 | 23.378 | −1.604 | 1 |
| Ecologist | 0.133 | 0.0466 | 0.1245 | 0.0896 | 0.1582 | 0.0259 | 0.1358 | 0.0861 | 77.760 | 26.983 | −1.341 | 3 |
| Special | 0.096 | 0.4137 | 0.1148 | 0.1670 | 0.2053 | 0.0009 | 0.1550 | 0.0312 | 74.457 | 26.892 | −1.045 | 4 |
| Committed | 0.107 | 0.2547 | 0.1384 | 0.0307 | 0.2426 | <0.0001 | 0.1560 | 0.0294 | 89.751 | 16.432 | −2.652 | 1 |
| Integrator | 0.109 | 0.2328 | 0.1307 | 0.0572 | 0.1643 | 0.0179 | 0.1460 | 0.0513 | 84.072 | 20.463 | −1.753 | 0 |
| Enthusiastic | 0.126 | 0.0790 | 0.1364 | 0.0365 | 0.1256 | 0.1372 | 0.0984 | 0.3695 | 86.335 | 19.170 | −1.960 | 0 |
| Responsable | 0.126 | 0.0790 | 0.1566 | 0.0056 | 0.1828 | 0.0052 | 0.0770 | 0.6237 | 88.597 | 17.085 | −2.373 | 2 |
| Comprehensive | 0.141 | 0.0237 | 0.1584 | 0.0046 | 0.1918 | 0.0027 | 0.2164 | 0.0004 | 88.439 | 16.456 | −2.227 | 3 |
| Generous | 0.107 | 0.2542 | 0.1511 | 0.0096 | 0.2373 | 0.0001 | 0.1063 | 0.2879 | 91.199 | 14.821 | −2.659 | 2 |
| Encouraging | 0.068 | 0.8465 | 0.1374 | 0.0336 | 0.1417 | 0.0643 | 0.1777 | 0.0075 | 85.520 | 20.851 | −2.148 | 1 |
| Kind | 0.124 | 0.0936 | 0.1594 | 0.0041 | 0.1761 | 0.0083 | 0.1460 | 0.0512 | 88.371 | 18.097 | −2.306 | 2 |
| Noble | 0.101 | 0.1726 | 0.1198 | 0.0482 | 0.1408 | 0.0326 | 0.1002 | 0.2175 | 87.511 | 18.532 | −2.024 | 0 |
| Respectful | 0.153 | 0.0079 | 0.1261 | 0.0802 | 0.1409 | 0.0669 | 0.1408 | 0.0672 | 88.032 | 17.129 | −2.319 | 1 |
| Optimistic | 0.120 | 0.1201 | 0.1235 | 0.0963 | 0.1894 | 0.0032 | 0.1696 | 0.0128 | 86.900 | 20.251 | −2.413 | 1 |
| Empathic | 0.142 | 0.0218 | 0.1351 | 0.0403 | 0.2359 | 0.0001 | 0.2060 | 0.0009 | 89.615 | 17.781 | −2.723 | 2 |
| Positive | 0.168 | 0.0003 | 0.1826 | <0.00001 | 0.1403 | 0.0336 | 0.1179 | 0.1048 | 89.186 | 17.070 | −2.615 | 2 |
| Helpful | 0.109 | 0.2366 | 0.1428 | 0.0209 | 0.2315 | 0.0001 | 0.1725 | 0.0106 | 90.475 | 16.822 | −2.820 | 1 |
| Tireless | 0.081 | 0.6684 | 0.1328 | 0.0486 | 0.0951 | 0.4060 | 0.1404 | 0.0687 | 65.860 | 29.212 | −0.587 | 3 |
| Modest | 0.108 | 0.2380 | 0.1385 | 0.0306 | 0.1002 | 0.3500 | 0.1169 | 0.1959 | 71.878 | 28.790 | −0.933 | 3 |
| Trustworthy | 0.092 | 0.4795 | 0.1674 | 0.0017 | 0.0979 | 0.3752 | 0.1188 | 0.1817 | 83.416 | 21.902 | −1.674 | 1 |
| Honest | 0.160 | 0.0038 | 0.136 | 0.0376 | 0.1521 | 0.0367 | 0.0886 | 0.4825 | 84.819 | 21.415 | −1.940 | 1 |
| Solidary | 0.105 | 0.1320 | 0.1607 | 0.0009 | 0.2315 | <0.0001 | 0.1410 | 0.0322 | 91.765 | 14.128 | −2.991 | 2 |
| Charismatic | 0.078 | 0.7195 | 0.1497 | 0.0111 | 0.109 | 0.2623 | 0.1401 | 0.0697 | 78.416 | 25.435 | −1.366 | 0 |
| Sensitive | 0.058 | 0.9329 | 0.0907 | 0.5067 | 0.2219 | 0.0002 | 0.1082 | 0.2700 | 82.579 | 21.658 | −1.624 | 1 |
| Equitable | 0.109 | 0.2291 | 0.123 | 0.0997 | 0.1919 | 0.0027 | 0.1456 | 0.0525 | 82.059 | 22.337 | −1.555 | 1 |
| Collaborative | 0.081 | 0.4427 | 0.1149 | 0.0695 | 0.2254 | 0.0001 | 0.1125 | 0.1332 | 91.018 | 15.448 | −3.115 | 1 |

*4.3. Statistical Analysis for Updating the SRBP Dimension*

In the first instance, a cluster analysis was carried out. This multivariate statistical analysis technique allows the adjectives to be classified, forming clusters (groups) that are as homogeneous as possible with each other based on their internal cohesion, and, in turn, heterogeneity among them based on the external isolation of the cluster. Subsequently, a factor analysis is carried out to investigate the presence of variables underlying the data set.

### 4.3.1. Cluster Analysis Results

Figure 2 allows observing the results of the cluster analysis, in which four groups of adjectives and two individual adjectives are identified.

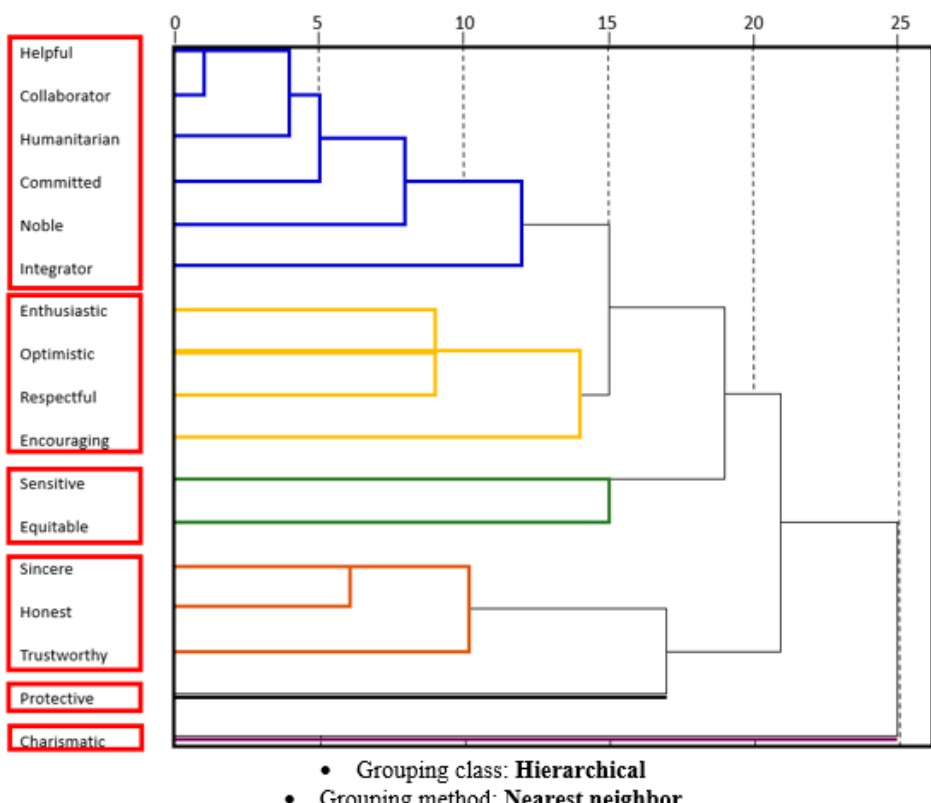

- Grouping class: **Hierarchical**
- Grouping method: **Nearest neighbor**
- Similarity: **Euclidean Distance**

**Figure 2.** Graphic representation of adjectives grouping.

### 4.3.2. Factor Analysis Results

The use of this statistical tool, called factor analysis, has facilitated in this study "the analysis of the interrelation patterns between variables, (...), classify and describe them (...)" [53], being able to improve the structure obtained after the cluster analysis previously carried out.

Table 4 shows that the factor analysis is done by using three rotation methods (varimax, quartimax and equamax, each one with Kaiser normalization), and two extraction methods (main components and generalized least squares). The crossing of these methods has shown six different scenarios, closely related to each other, which allow us to observe very interesting patterns for discussion.

**Table 4.** Factor loads of factorial scenarios (three factors).

|  | | Rotation Method: Varimax with Kaiser Normalization | | | Rotation Method: Quartimax with Kaiser Normalization | | | Rotation Method: Equamax with Kaiser Normalization | | |
|---|---|---|---|---|---|---|---|---|---|---|
|  | | Factor | Factor | Factor | Factor | Factor | Factor | Factor | Factor | Factor |
|  | Adjective | I | II | III | I | II | III | I | II | III |
| Extraction method: main components | Sincere | 0.734 | 0.171 | 0.173 | 0.771 | −0.059 | −0.001 | 0.722 | 0.191 | 0.203 |
|  | Humanitarian | −0.014 | 0.400 | 0.670 | 0.258 | 0.404 | 0.616 | −0.051 | 0.385 | 0.677 |
|  | Protective | 0.560 | 0.078 | 0.292 | 0.609 | −0.09 | 0.16 | 0.546 | 0.090 | 0.314 |
|  | Committed | 0.183 | 0.702 | 0.190 | 0.420 | 0.617 | 0.078 | 0.153 | 0.704 | 0.212 |
|  | Integrator | 0.300 | 0.563 | 0.263 | 0.504 | 0.451 | 0.138 | 0.272 | 0.567 | 0.286 |
|  | Enthusiastic | 0.455 | 0.619 | −0.004 | 0.603 | 0.448 | −0.16 | 0.434 | 0.633 | 0.025 |
|  | Encouraging | 0.310 | 0.143 | 0.610 | 0.470 | 0.058 | 0.514 | 0.284 | 0.140 | 0.623 |
|  | Noble | 0.248 | 0.271 | 0.724 | 0.475 | 0.202 | 0.627 | 0.213 | 0.262 | 0.738 |
|  | Respectful | 0.617 | 0.331 | 0.180 | 0.711 | 0.13 | 0.014 | 0.599 | 0.347 | 0.209 |
|  | Optimistic | 0.532 | 0.491 | 0.060 | 0.652 | 0.304 | −0.101 | 0.513 | 0.506 | 0.089 |
|  | Helpful | 0.143 | 0.708 | 0.317 | 0.414 | 0.638 | 0.209 | 0.108 | 0.705 | 0.337 |
|  | Trustworthy | 0.761 | 0.201 | 0.177 | 0.806 | −0.038 | −0.006 | 0.747 | 0.222 | 0.208 |
|  | Honest | 0.770 | 0.155 | 0.173 | 0.800 | −0.086 | −0.008 | 0.758 | 0.176 | 0.203 |
|  | Charismatic | 0.625 | 0.200 | 0.148 | 0.672 | 0.002 | −0.006 | 0.612 | 0.217 | 0.175 |
|  | Sensitive | 0.284 | 0.114 | 0.598 | 0.433 | 0.038 | 0.511 | 0.259 | 0.109 | 0.610 |
|  | Equitable | 0.488 | 0.289 | 0.325 | 0.612 | 0.134 | 0.186 | 0.466 | 0.298 | 0.349 |
|  | Collaborative | 0.175 | 0.693 | 0.358 | 0.448 | 0.616 | 0.244 | 0.139 | 0.690 | 0.379 |
| Extraction method: generalized least squares | Sincere | 0.704 | 0.187 | 0.185 | 0.672 | −0.333 | −0.049 | 0.691 | 0.213 | 0.204 |
|  | Humanitarian | 0.106 | 0.215 | 0.657 | 0.464 | 0.154 | 0.499 | 0.085 | 0.194 | 0.666 |
|  | Protective | 0.467 | 0.205 | 0.188 | 0.521 | −0.155 | −0.001 | 0.453 | 0.220 | 0.204 |
|  | Committed | 0.224 | 0.46 | 0.368 | 0.579 | 0.215 | 0.125 | 0.195 | 0.456 | 0.389 |
|  | Integrator | 0.291 | 0.475 | 0.318 | 0.615 | 0.174 | 0.06 | 0.262 | 0.476 | 0.341 |
|  | Enthusiastic | 0.357 | 0.645 | 0.085 | 0.671 | 0.223 | −0.228 | 0.324 | 0.658 | 0.115 |
|  | Encouraging | 0.281 | 0.248 | 0.437 | 0.518 | 0.031 | 0.25 | 0.262 | 0.244 | 0.451 |
|  | Noble | 0.309 | 0.134 | 0.748 | 0.592 | −0.034 | 0.568 | 0.291 | 0.120 | 0.758 |
|  | Respectful | 0.531 | 0.363 | 0.21 | 0.67 | −0.085 | −0.051 | 0.510 | 0.380 | 0.233 |
|  | Optimistic | 0.446 | 0.491 | 0.144 | 0.662 | 0.058 | −0.138 | 0.419 | 0.506 | 0.170 |
|  | Helpful | 0.157 | 0.574 | 0.431 | 0.627 | 0.349 | 0.156 | 0.123 | 0.564 | 0.455 |
|  | Trustworthy | 0.733 | 0.227 | 0.177 | 0.713 | −0.326 | −0.077 | 0.719 | 0.255 | 0.198 |
|  | Honest | 0.806 | 0.124 | 0.183 | 0.703 | −0.449 | −0.051 | 0.796 | 0.155 | 0.201 |
|  | Charismatic | 0.503 | 0.286 | 0.156 | 0.582 | −0.126 | −0.068 | 0.486 | 0.304 | 0.175 |
|  | Sensitive | 0.251 | 0.29 | 0.33 | 0.48 | 0.07 | 0.144 | 0.232 | 0.289 | 0.345 |
|  | Equitable | 0.421 | 0.386 | 0.237 | 0.618 | 0.011 | −0.011 | 0.398 | 0.396 | 0.258 |
|  | Collaborative | 0.193 | 0.56 | 0.46 | 0.655 | 0.318 | 0.18 | 0.158 | 0.551 | 0.484 |

By making a detailed review of the factor loads of each scenario, three clear factors can be determined as shown in Table 3. For the elaboration of each suggested factor, a count of the times that each variable is more correlated with that factor is made. In this way, the variable is linked to that factor, that is, if the adjective *committed* was more correlated to factor I (once) and five times more correlated to factor II in the suggested scenario, it is then linked to factor II. Each of the 17 adjectives were reviewed in each of the scenarios in a meticulous and detailed way to determine their association, in order to obtain an ideal structure for the dimension.

Table 5 suggests 3 factors conformed as follows:

1. Factor I: sincere, protective, respectful, optimistic, trustworthy, honest, charismatic and equitable;
2. Factor II: committed, integrator, enthusiastic, helpful and collaborative;
3. Factor III: humanitarian, encouraging, noble, sensitive.

**Table 5.** Suggested factors from factorial scenarios (three factors).

| | Suggested Factor | | |
|---|---|---|---|
| Adjective | Factor I | Factor II | Factor III |
| Sincere | 6 | 0 | 0 |
| Humanitarian | 0 | 0 | 6 |
| Protective | 6 | 0 | 0 |
| Committed | 1 | 5 | 0 |
| Integrator | 2 | 4 | 0 |
| Enthusiastic | 2 | 4 | 0 |
| Encouraging | 1 | 0 | 5 |
| Noble | 1 | 0 | 5 |
| Respectful | 6 | 0 | 0 |
| Optimistic | 4 | 2 | 0 |
| Helpful | 1 | 5 | 0 |
| Trustworthy | 6 | 0 | 0 |
| Honest | 6 | 0 | 0 |
| Charismatic | 6 | 0 | 0 |
| Sensitive | 1 | 0 | 5 |
| Equitable | 6 | 0 | 0 |
| Collaborative | 1 | 5 | 0 |

This would allow inferring that the SRBP dimension would be made up of 3 attributes and 17 traits, something similar to [44], who in his proposal determines that this dimension is composed of 3 attributes and 15 traits.

The final configuration of the dimension this work proposes is not only done from a quantitative perspective, but also aims at understanding socially responsible behavior in detail, and that is precisely why defining the dimension will contribute to preparing appropriate narratives for brands and will help to optimize the relationship process between the brand and audiences, based on strategic, empathic and much more humanized communication. In the discussion section, the final structure will be presented, detailing each of the attributes and explaining the associations of their traits as descriptors of the socially responsible personality type.

## 5. Discussion

Several reviews of literature developed by this work team show a gap in the structuring of a socially responsible brand. Likewise, the most current research is the verification of a postulate from the beginning of the 21st century, which affirms that corporate social responsibility is a fundamental pillar of the management of reputation of organizations and, in turn, it contributes to the creation of strong brands.

Taking into account the current market conditions, it is required that contemporary brands can be more long-lived, that they contribute to influence distribution channels, help expand sales outside national markets, serve to attract and retain high-level personnel, and obviously contribute to increasing company profits [57], but also that they become the central axis of the relationship process with stakeholders. Therefore, understanding that today's consumers are increasingly interested in environmental and social issues, it is required that the structuring of a contemporary brand be developed from the same perspective of the citizens/consumers.

Concepts such as equity brands and/or brands with purpose are occupying the front pages of the media, are used as campaign slogans and are popularized as hashtags on

social networks. For this, the academy requires a position on these concepts that today are present in the meeting rooms of company boards of directors. A relevant element of brand management is its personality. This management tool has allowed company managers to clearly identify the attributes and traits with which they are perceived by their public, which contributes to the reformulation of a much more focused brand identity.

That is why the constant updating and adaptation of the concept becomes an academic necessity since society lives a continuous evolution that requires an active academy. To achieve a strong brand in the long term, a process of constant monitoring of the brand is required, not simply from a purely visual perspective but from a more holistic perspective, understanding that the brand is a market player and, as such, it must be evaluated by the perception of the consumers, competitors, and, in general, of all the actors related to the brand.

Therefore, it is necessary to continue perfecting the construct called brand personality, not only because it is necessary for the academy to enter an essential element for in-depth knowledge of the brand as a social agent, but also because it is necessary to provide the industry with tools of brand management updated and consistent with current contexts. According to [19], brand personality influences the attitudes and cognitive associations that audiences have in relation to brands, and it also generates emotions in the consumers. On the other hand, it encourages self-expression and association of individuals in relation to a brand. It is also a key element to stimulate differentiation, contributing to the processing of information issued by brands and, especially, increasing levels of trust and loyalty, as well as influencing preferences and use by consumers.

As stated by [31] (p. 30),

*"Brand personality is relevant to be studied and analyzed, since it has been shown that individuals have related human characteristics of emotionality and personality to brands, in order to express their experience and opinion of them [26]; Haigood [22] as consumers seek to identify and share their values with brands [58]".*

For this, the constant updating and adaptation of the concept becomes an academic necessity since society is living a continuous evolution, which requires an active academy.

As society currently lives in the era of sustainability, studying the brand management at this time is something supremely convenient, not only for the simple updating of concepts and tools but because the role of brands in this era has been substantially modified.

According to [59], brands are powerful instruments of change today. They also state that brands are now closely related to their consumers since they are deeply incorporated into their daily lives. This strong relationship that these authors propose is materialized in the constant search of individuals for brands that represent their way of thinking, feeling and being, as well as adapt to the image they want to project.

Thus, according to [59] (p. 78) "brands that respect the environment are an inevitable element of the sustainable marketing strategy and the concept of sustainability, since their application requires changes that will have an effect on the multitude and not on individuals.", but they clarify that at present "regardless of the positive opinion on the socially responsible practice in the market, the attitude and behavior gap is very present among consumers, which makes the ecological consumer segment just a niche" (p. 78).

Therefore, the structuring of brands with socially responsible features becomes a priority need for organizations wanting to compete given the current market conditions. Grubor and Milovanov [59] state that "the adoption of sustainable attitudes and behaviors through the use of sustainable brands has the power to initiate deeper changes in people's lives" (p. 79), evidently contributing to the *Triple Bottom Line* of organizations and ensuring a balance between the following three edges responsible for sustainable development: companies, society, and consumers.

### 5.1. Selection of Adjectives

As mentioned in the previous section, 17 adjectives were selected to be included in the final structure of the dimension. Unlike [44], the selection made in the Peruvian context in

times of pandemic excluded the following adjectives: kind, ecological, generous, positive, responsible, and solidary, compared to the 2017 list.

On the other hand, it included, unlike the first proposal of an SRBP dimension, the following adjectives: enthusiastic, honest, integrator, optimistic, protective, respectful, sensitive, and sincere. The inclusion and exclusion criteria have already been exposed previously. On the reasons why the assessment showed this list, it can be inferred that they were due to the cultural context and the moment of taking the information (in the midst of the COVID-19 pandemic).

It is important to mention that the mean valuation of the adjectives was 83.32, much higher than that obtained when the first-dimension structure proposal was made in 2017. This is a situation that cannot be explained from the data collected or with the analysis techniques used, and this could be an effect of the moment of data collection, since the world population is much more sensitive.

### 5.2. Structuring and Updating the SRBP Dimension

We can find certain similarities between the results obtained in the two stages of the factor analysis. Two nuclear factors of the structure have been detected; in Table 6 these factors can be observed and also the groups of adjectives that have a high relationship can be identified. In that table, it is observed that the adjective *enthusiastic* is not found, and this does not indicate that it has been excluded from the structure, but it presents a very volatile behavior, that is, depending on the statistical technique used it generates relationships with different groups of adjectives, so its final location will be determined by a very precise situation for that case.

**Table 6.** Nuclear factors of the structuring of the dimension.

| NUCLEAR FACTOR I | NUCLEAR FACTOR II |
|:---:|:---:|
| Sincere | Committed |
| Trustworthy | Integrator |
| Honest | Helpful |
| Respectful | Collaborative |
| Optimistic | Humanitarian |
| Protective | Noble |
| Charismatic | Encouraging |
| Equitable | Sensitive |

After the detailed review process of the results, it can be determined that the dimension is structured by two factors, from now on called attributes, described by 17 adjectives, from now on called traits. Figure 3 shows the final structure proposed as an SRBP dimension.

This structure has three levels, keeping the format used by [44] in which the author determines that the dimension has a virtue described by two attributes that, in turn, contain 17 traits. According to Mayorga, a virtue "is a superior disposition of personality that has real, independent, individual existence, identified through a group of properties called attributes". [44].

Likewise, he defines attributes as the "permanent and essential element of the personality, identified through a set of distinctive peculiarities called traits." [44]; finally, he defines traits as "the singular character of a person that identifies him/her, makes him/her different and unmistakable." [44].

It is important to mention that when comparing the structure of the dimension proposed by [44] and the structure proposed herein, it is identified that the attribute called altruistic has some similar traits, such as the following: humanitarian, noble, helpful, and collaborative, turning them into a group of representative adjectives of that attribute. This allows us to conclude that altruism can become a representative attribute of a socially responsible brand.

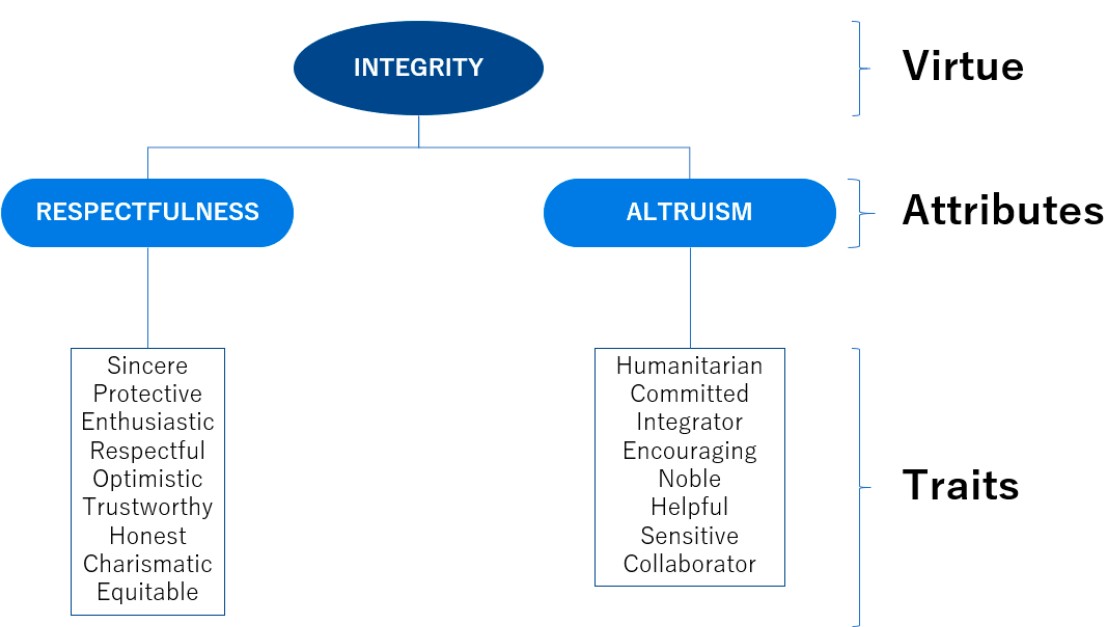

**Figure 3.** Updated structure of the SRBP dimension in times of COVID-19.

On the other hand, the attribute respectful is similar to the attribute trustworthy proposed by [44]. It can be said that this attribute is contained by respectful, since they share the trustworthy, charismatic and equitable traits, which are a representative part of the attribute trustworthy proposed by Mayorga.

*5.3. Definition of SRBP Dimension in Times of COVID-19*

The SRBP dimension is defined by a virtue called integrity, which is defined as "quality of integrity" [60] (upright: two adjectives said of a person are straight, prove, faultless. [60]). Therefore, the integrity of an individual is characterized by his/her severity with himself/herself and with others, in compliance with moral and conduct standards, besides being fair, correct, honest, and faultless.

According to [61], integrity "favors solid interpersonal relationships and helps build the common good." (p. 12) They also affirm that the effort to achieve it "produces undeniable positive effects in the lives of people in general" (p. 12). The authors state that "betting on integrity is preparing to harmoniously reconcile one's own good and the common good" (p. 12). The integrity of a person makes his/her word have value, since it offers guarantees and that the result of his/her actions puts their own interests aside.

Integrity is the cornerstone of reliability, since it is a virtue based on compliance, that is, it not only does what it says but it does it beyond its interests and, above all, seeking to benefit the community. Upright people are, in turn, frank and transparent, traits that greatly favor communication, since integrity favors solidary and lasting relationships, besides contributing by weaving interpersonal networks based on trust. According to [61],

> "On the basis of integrity, the reputation of the person is built and, by reflex, also that of institutions, when these are led according to the criterion of integrity. A good reputation consists of enjoying the recognition of others, based on trust, the rectitude of its intentions, and backed by a career of transparency and honesty in its actions." (p. 13).

An upright brand is characterized by being altruistic and respectful. According to the new lexicographical thesaurus of the Spanish language of the RAE, altruista was defined in 1917 by Alemany and Bolufer as the following:

> "Self-denial, benevolence for the benefit of others; fulfillment of moral duties in favor of others. Defense of social equality by feeling of justice. Denunciation of all kinds of advantages and privileges for considering that social assets belong or should belong, equally to all members of society" [60].

Therefore, an altruistic brand is distinguished by being benevolent, selfless, fair and, above all, interested in others, without putting its interests first. According to the structure developed in this work, an altruistic brand is above all humanitarian, committed, integrator, encouraging, noble, helpful, sensitive and collaborative.

On the other hand, an upright brand is characterized by being respectful. Since the first dictionaries of the Spanish language, the quality of respectful is described as "(adj.) What causes or moves to veneration and respect. That one who observes veneration, courtesy and respect." [60]. Therefore, if the brand is respectful it will be a courteous, considerate, attentive, prudent and moderate brand. Clearly it is a brand that thinks of others from a position of service, help, and collaboration, but always being sincere and honest, and above all, equanimous and fair.

So, a respectful brand can be described from the traits that define the structure developed in this work, such as the following: sincere, protective, enthusiastic, respectful, optimistic, trustworthy, honest, charismatic, and equitable.

### 5.4. Strategic Management of the Socially Responsible Brand

Currently, customers increasingly demand responsible behavior from brands and their manufacturers, placing it as a determining factor when choosing brands or products/services. Therefore, engaging with audiences from integrity is not optional for brands that intend to compete in the current market under the conditions experienced. Thus, brands must develop communication plans focused on transmitting a message of integrity to their consumers, adjusting their strategic management plans for their brands and optimizing their general behavior within their environment.

The era of sustainability requires brands to develop narratives based on altruism and respect, allowing the public to identify in them an attitude towards commitment to the environment, the norms of society, and the general progress of humanity.

According to Gabriela Álvarez, "sustainability (...) is about collaborating, learning, creating, implementing, evaluating and constantly evolving" [59]. Therefore, corporate communication, which is in charge of leading the relationship with the audiences, must understand that the construction of narratives, messages, and, obviously, the brand management, is not an individual but a collective exercise. It must be developed from the perspective of co-creation, without forgetting that the center of communicative management is the public, and, in the same way, the core of the marketing strategy is the consumer, an increasingly aware, informed, and active consumer, a fact that organizations cannot ignore when determining their strategic plans.

According to [59], "sustainability should be considered as a process integrated in every process of a company, in order to achieve the holistic adoption of sustainable principles". Therefore, the building of a socially responsible brand is not an external tactic, but should be considered a central axis of strategic business management. All the actions developed in the company must respond to these proposed virtues and focus the organization management in an increasingly convulsive, critical and unstable context, which requires actions aimed at achieving a sustainable development of society.

Today's companies must change their corporate values into virtues such as sensitivity [44], resilience, and the proposal in this research, integrity. Organizations must understand, not only the generational change that is being experienced, but also the global interests of humanity, in order to direct their organizations, achieving a current position and becoming more competitive.

This dimension arises from a young, irreverent vision, eager for changes, interconnected, global, and, above all, concerned about a better future. Thus, communicational management must interpret the results of this work as a call for change and modernization in order to establish long-term links with the public.

Today's brands pursue interests, in many cases, far from the interests of their audiences, so integrating the SRBP dimension into business dynamics will not only allow them to optimize their relationships with the different stakeholders, but also to establish a modern

identity committed to the needs of the environment, thus being able to achieve long-term competitive advantages. Then, the following statement of [59] (p. 79) becomes relevant: "joint work is a new mantra that puts stakeholders in a position to develop a common language, trust, and a shared vision with all the partners". Sustainability management cannot be isolated and this is where the brand takes on its leading role as a natural element of integration and relationship between companies.

## 6. Conclusions

Since 1997, when the brand personality model was first introduced, the academy has been interested in clearly defining the most appropriate attributes and traits to describe a brand, but few contributions have been made in relation to the socially responsible brand personality. Therefore, this research is a contribution not only to the disciplines related to marketing, advertising and corporate communication, but also to the disciplines interested in the theoretical formulation of corporate social responsibility, since the identification of the previously exposed structure contributes to the identification of mental associations that help the brand imagery and contribute to the establishment of a corporate vision and mission with social purpose. Therefore, under the current social, environmental and economic conditions, the approach to a socially responsible brand personality structure becomes relevant for the business and academic world.

The results of this work allow both the academic world and the industry to understand their current audiences from a strategic communication and marketing perspective. The context of the pandemic is changing the way that the public and companies interact, and time cannot pass without describing this process in the midst of the new context.

1.  From the results obtained in this research, it can be supposed that the current context (COVID-19 pandemic) makes a change in the assessment that individuals make of adjectives. Out of the 30 initial adjectives, 24 obtained a score higher than 80/100. In the 2017 study, only seven adjectives obtained a high score;

2.  This could lead to deduce that the assessment made in the Peruvian context, in pandemic times, was somewhat more "benevolent" than in a context without a global health, social, and economic crisis, in which people are demanding a change in the attitude of brands;

3.  After making a comparison between the results of this work and the 2017 proposal, seven adjectives can be identified (disinterested, charitable, special, comprehensive, empathic, tireless and modest) that have not been included in either of the two lists developed so far. It could be deduced that they are adjectives that do not describe a socially responsible personality. For future replications of this work, they should be included in order to corroborate their definitive exclusion from the list of possible constitutive features of the dimension;

4.  Likewise, nine adjectives (humanitarian, committed, encouraging, noble, helpful, trustworthy, charismatic, equitable, and collaborative) were identified in the final structures of the two studies, so that it can be said that these ones are an essential constitutive part of the structure of such a dimension;

5.  One of the most significant conclusions of this study is the permanence of altruism as an attribute in the dimension structure since, although this attribute is made up of different features in the two studies, it emerges as a statistical factor. In both the cases, this attribute includes humanitarian, noble, helpful, and collaborative as descriptive traits, a fact that indicates the relevance of the attribute and the relationship between these four traits, and that this deserves further investigation in the future.

In convulsive times like the one the world is currently experiencing, it is necessary to create mechanisms that contribute to the strategical management of the relationship with the public, adapted to their needs. The building of brands and communicational and relational management adapted to the context, and, above all, focused on the deep understanding of the public, require that brands adapt to their management a dimension

of their personality in line with sustainability and CSR. Therefore, the result of this research stands as a fundamental tool for business relations today.

Several authors claim that contemporary consumers have increasingly become astute about CSR issues and activities, while being more perceptive about specific CSR practices in companies. This research, therefore, provides the academy with a framework to evaluate the socially responsible image of brands from an anthropomorphic perspective.

In this sense, some academics state that CSR actions help soften the crisis situations that organizations experience, since they promote positive consumption or purchase behaviors on individuals. Therefore, communication management based on a SRBP dimension allows organizations to better deal with crises.

Currently, it is stated that strong brands are considered as an especially important engine of change, since they are erected as bulwarks of sustainable behaviors of both companies and consumers. At this time, sustainability must be recognized as a relevant concept in the business world, so it is necessary to adapt internal culture and brand image in that direction.

Researchers from various disciplines related to the business world point out that modifications are required in the marketing policy and culture of modern organizations, so that a brand management that adopts socially responsible attributes and traits becomes an especially important line of action for navigating sustainable development.

Therefore, the findings of this work contribute to the management of brand identity and image, since, during a globalized, interconnected, and, above all, virtualized context, the need to determine clear equity brands, communicated to all the stakeholders of the organizations, should be a priority for their senior executives.

This work is the starting point of several studies aimed at determining not only the level of perception of a socially responsible brand by the public, but also the effects of a communication management that interprets this structure in favor of the building of a strong brand in terms of sustainability. Thanks to the structure proposed here, it is possible to develop scales to measure the perception of corporate social responsibility of an industry, a category of products or an organization and its brands. Likewise, this work also contributes to the theoretical conceptualization of CSR, since it provides adjectives of daily use with which citizens interpret social responsibility, simplifying the understanding of this theoretical construct and bringing it closer in one way or another to the community in general.

In this new normal, where humanity currently operates, the understanding of a socially responsible company has drastically changed. Consumers and different publics of interest demand brands to not only to show responsible behavior in terms of the environment, but also now they ask them to be much more critical and active companies in terms of social needs. Therefore, this adaptation of the personality structure of a brand with social responsibility is an important step for the general understanding of the new consumer.

*6.1. Theoretical and Practical Implications*

From the perspective of brand equity, organizations need to manage their brand as relevant assets within the financial structure of organizations and as a tool for the relationship with their public. Therefore, any theoretical contribution to branding becomes a step towards the modernization of brands in a convulsive and changing context.

Various sectors that work in the building of brands, such as the advertising and public relations industry, as well as corporate communication, marketing and social responsibility consultancies, require structured models to identify the mental structures of the market that their brands or those of their customers represent. For this reason, the model proposed by Aaker has been of great importance in the industry, since it has allowed the interpretation of brands from the perspective of the theory of anthropomorphism and attribution. This contribution has helped company managers not only to identify their objectives with higher quality, but it has also allowed them to better understand and interpret the brands they manage.

This work, which is the result of several years of research in relation to brand personality and its strategic management, provides a new vision of brand personality and, above all, a perspective of the brand as a construct. The results of this work are of great importance for the industry, since they are the reflection of a society interested in a change in attitude by companies, and a demand towards more socially and environmentally responsible behavior. The structure that this study revealed helps to measure the perception of brands by their market segments from a sustainable perspective, but also updated, since this study was carried out during the first peaks of the COVID-19 pandemic.

On the other hand, this structure contributes to the creation of more effective messages, and, above all, ones closer to the motivations of current audiences. It also allows the integrated marketing communication strategy to be focused on sustainability, which allows brands to modernize and better connect with the young public, which is becoming a market with great potential.

Finally, for companies and their managers, this structure allows setting clear sustainability goals and designing balanced scorecards that interpret their position from the perspective of corporate social responsibility, which is especially important for contemporary managers who see, in sustainable development goals, a roadmap for their management.

### 6.2. Limitations and Future Research

This work was developed during a complex public health context, accompanied by a difficult economic and political situation worldwide. Therefore, to set limitations to the research would be redundant. Clearly, this work presented a difficulty in data collection, and, in turn, it can be inferred that the assessment of each of the proposed items was influenced by the context, since this research was carried out in the middle of the first peak of the pandemic.

However, this limitation is a possibility for future research, in which replications of this study could be carried out to compare the assessment of each of the variables included in this study, and thus theorize about the modulating effect of the crisis.

It is also important to replicate this study in different social, economic, and cultural contexts, since with the results obtained it will be possible to formalize a global structure of the socially responsible brand personality, a model that will contribute to the improvement in business relationship processes and will contribute to the management of corporate sustainability.

At the same time, it opens the possibility of validating the structure proposed by the academy in the industry, based on empirical experimentation that provides data allowing the formalization of the model, and, in turn, contributes to its consolidation. It would be of great value for the consolidation of the model that various disciplines of economic, management, and social sciences in general, help from their perspective to the configuration of a much more holistic model.

**Author Contributions:** Conceptualization, E.E.G.-S.; methodology, J.M.G.; software, J.M.G.; validation, J.M.G and E.E.G.-S; formal analysis, J.M.G.; investigation, E.E.G.-S.; resources, J.M.G.; data curation, J.M.G.; writing—original draft preparation, E.E.G.-S.; writing—review and editing, J.M.G.; visualization, J.M.G.; supervision, J.M.G.; project administration, E.E.G.-S. All authors have read and agreed to the published version of the manuscript.

**Funding:** This research received no external funding.

**Institutional Review Board Statement:** Ethical review and approval were waived for this study, due to the data are completely anonymous and informed consent was obtained at the time of original data collection.

**Informed Consent Statement:** Informed consent was obtained from all subjects involved in the study.

**Data Availability Statement:** Not applicable.

**Acknowledgments:** This paper was possible thanks to the financial and logistical support of the Universidad Privada del Norte and the Universidad Nacional Tecnológica de Lima Sur de Perú, together with the Universidad Autonoma de Occidente de Cali-Colombia.

**Conflicts of Interest:** The authors declare no conflict of interest.

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
