# Peer review of "Brand Personality as a Consistency Factor in the Pillars of CSR Management in the New Normal"

_2199-8531, doi:10.3390/joitmc7020134_

Round 1

Reviewer 1 Report

The abstract should be improved and clearly identify the objective of this study and the contributions of the study should be included.

The introduction should include the gaps that justify the relevance of this study, before the objectives.  A paragraph with the structure of the article should be placed at the end of point 1.

The points 1.1. and remaining should be modified to :
2. Literature review
2.1. 
2.2. etc.

In the material and data collection part, the authors should sisitematize in tables the sample used and its characterization. Explain how the information was obtained. It is not clear!!!

finally the whole conclusion should be revised. It should not include bibliographical references, it should have a summary of the conclusions in view of the results obtained, it should include the contributions of this study and its implications for theory and practice; how this study contributed to the advance of knowledge of the model used; the limitations and the future research agenda.

Reviewer 2 Report

Dear Author(s),

This is an interesting paper on a topical research area. While the topic is clearly aligned with our journal’s aim and scope, there are several concerns regarding the practical relevance and conceptual foundation for your study, as well as the research design, that makes it difficult to determine if or how your study makes a significant contribution to previous research in this area.

It is recommended that the authors address the following comments and suggestions.

ABSTRACT

Please add a clear research objective in the abstract. As this should be the starting sentence for the abstract, you have to try to sharpen its focus.

Overall, I would make the abstract sharper, answering to the following questions:

- What is the problem?

- What have you done?

- What is the main contribution of the paper?

INTRODUCTION

The motivation for this study in the introduction is not strong enough. Be clearer about why we should care about this topic and what the gap is in the literature.

My chief concern is that the originality of the paper is not clearly explained nor in the abstract, nor in the introduction.

In fact, the introduction section is quite confusing. Please rewrite it following the items below:

- Establish the importance of research.

- Establish a theory based gap.

- Explain contribution.

- Present the overall paper structure.

LITERATURE REVIEW

The conceptual background section needs to be clear and more related to the research framework

METHODOLOGY

The methodology section mainly is the paper’s argument built on an appropriate base of theory, concepts, or other ideas. The method and methodology employed should be explained and correctly interpreted taking into account the reason for choosing the current methodology and adding information about past studies that applied the same methods in similar areas. Please, pay attention to these issues.

DISCUSSION

There is a need for more discussion on how achieved findings can be connected to the previous conceptual background. Please link the discussion of the research problem with the highlighted gaps in the conceptual background. In fact, it is expected to use and criticize any relevant literature (in the literature review section) in order to reach the main purpose of the study to explain how the paper succeeded in enriching the development of the selected topic as well as to explore different views.

Please, make some linkage between the paragraphs.

There are some typos, please make sure you proofread the paper

THEORETICAL AND PRACTICAL IMPLICATIONS

To begin, the justification for your study from an industry/practical and social standpoint was lacking. This was a salient oversight on your part.

You should try to answer to the following questions: What kinds of objective evidence can you offer that would make industry leaders sit up and pay attention to your study? What makes this topic a big deal right now, and perhaps in the immediate future?

Answers to these and related questions will help make a much stronger case for pursuing this line of inquiry.

Additionally, the theoretical implications for your study were underdeveloped. In particular, you did not clearly articulate how your study advances that which we already know. 

CONCLUSIONS

I encourage you to clearly and cogently explain final section: (1) how the focal study addresses an important priority; (2) how the topic is connected to existing theory; (3) what we already know about practice and theory; (4) what specifically we do not know; (5) why we need to know what we do not know; and (6) how this study or inquiry will help close the practical and theoretical gaps between what we know and do not know.

Best Regards

Round 2

Reviewer 1 Report

Good Morning
All suggestions were accepted and included by the authors.

Reviewer 2 Report

I'm happy with the revisions provided by the authors.

Best Regards